

# Metagenome-wide characterization of shared antimicrobial resistance genes in sympatric people and lemurs in rural Madagascar

Brooke M. Talbot[1,2], Julie A. Clennon[3], Miarintsoa Fara Nantenaina Rakotoarison[4], Lydia Rautman[3], Sarah Durry[5], Leo J. Ragazzo[3], Patricia C. Wright[4,6], Thomas R. Gillespie[1,3,4,5] and Timothy D. Read[1,2]

[1] Program in Population Biology, Ecology, and Evolution, Emory University, Atlanta, GA, United States of America
[2] Division of Infectious Diseases, School of Medicine, Emory University, Atlanta, GA, United States of America
[3] Department of Environmental Sciences, Emory University, Atlanta, GA, United States of America
[4] Centre ValBio, Ranomafana, Madagascar
[5] Department of Environmental Health, Rollins School of Public Health, Emory University, Atlanta, GA, United States of America
[6] Institute for the Conservation of Tropical Ecosystems, State University of New York at Stony Brook, Stony Brook, NY, United States of America

Corresponding authors
Thomas R. Gillespie,
thomas.gillespie@emory.edu
Timothy D. Read, tread@emory.edu

## ABSTRACT

**Background**. Tracking the spread of antibiotic resistant bacteria is critical to reduce global morbidity and mortality associated with human and animal infections. There is a need to understand the role that wild animals in maintenance and transfer of antibiotic resistance genes (ARGs).

**Methods**. This study used metagenomics to identify and compare the abundance of bacterial species and ARGs detected in the gut microbiomes from sympatric humans and wild mouse lemurs in a forest-dominated, roadless region of Madagascar near Ranomafana National Park. We examined the contribution of human geographic location toward differences in ARG abundance and compared the genomic similarity of ARGs between host source microbiomes.

**Results**. Alpha and beta diversity of species and ARGs between host sources were distinct but maintained a similar number of detectable ARG alleles. Humans were differentially more abundant for four distinct tetracycline resistance-associated genes compared to lemurs. There was no significant difference in human ARG diversity from different locations. Human and lemur microbiomes shared 14 distinct ARGs with highly conserved in nucleotide identity. Synteny of ARG-associated assemblies revealed a distinct multidrug-resistant gene cassette carrying *dfrA1* and *aadA1* present in human and lemur microbiomes without evidence of geographic overlap, suggesting that these resistance genes could be widespread in this ecosystem. Further investigation into intermediary processes that maintain drug-resistant bacteria in wildlife settings is needed.

## INTRODUCTION

The global estimated number of human deaths attributed to antibiotic-resistance among bacterial infections in 2019 alone was 1.27 million, with Sub-Saharan African countries experiencing the highest proportion of the burden (*Antimicrobial Resistance Collaborators, 2022*). Antimicrobial resistance in bacteria is a heterogeneous problem, with multiple organisms, biological mechanisms, and anthropogenic activities contributing to its presence and spread. Pathogen spread is known to play a major role in antimicrobial resistance gene (ARG) distribution, with evidence of enteric infections among symptomatic humans and animals having less antibiotic susceptibility compared to asymptomatic individuals (*Parisi et al., 2020*). Bacteria can acquire antibiotic resistance through *de novo* mutations, but they may also acquire resistance through horizontal gene transfer on mobile genetic elements (MGEs). MGE movement through a bacterial community depends on the species present, as MGE sharing can be restricted by species compatibility and host range (*Cury et al., 2018*; *Jiang et al., 2019*; *Moller et al., 2021*), but the presence of ARGs and their transference into closely related species can facilitate epidemic spread of pathogens (*Baker et al., 2018*). Nearly all bacterial pathogens associated with infectious diseases have been found to contain antimicrobial resistance genes, so it becomes imperative to capture the extent to which illness in an area may drive antibiotic resistance.

Although reducing antimicrobial resistant infections in humans and animals is a global priority, there remain major gaps in measurements of the prevalence of antibiotic resistant organisms across species and region. Knowledge of transmission dynamics and prevalence of community-acquired antimicrobial resistant species shared between overlapping humans and animals is limited. Detecting the distribution and diversity of specific antimicrobial resistant genes (ARGs) within and between human and animal microbiomes can further identify potential spillover events. Although antibiotics are lifesaving during some infections, agricultural and medical overuse of antibiotics contribute to the current rise of resistant organisms in human and animal populations (*Nadimpalli et al., 2018*). Further, and consequently, domestic animals, peri-domestic rodents, and wildlife all harbor ARGs, and each group can act uniquely as a sentinel for emerging or increased spread of antibiotic resistance (*Aivelo, Laakkonen & Jernvall, 2016*; *Gwenzi et al., 2021*; *Peng et al., 2022*; *Huang et al., 2022*). Comparisons of resistomes are well documented between human and agricultural animals (*Bloomfield et al., 2022*; *Huang et al., 2022*), agricultural soil (*Fang et al., 2023*), and in wastewater (*Munk et al., 2022*), showing widespread ARG diversity that is geographically specific. A lesser focus has been on comparative studies of ARGs in wildlife animals overlapping with human communities.

In this paper, we examined human and brown mouse lemur gut microbiomes to investigate the extent of ARG sharing between humans and wildlife in rural Madagascar where there are opportunities for humans and lemur spatial overlap. Whether shared environment could be enough to result in shared microbiomes/resistomes is of interest, given that in Madagascar lemur species exist across a gradient of human-transformed space, from undisturbed wild to being kept as pets in some households. The gradient of lifestyle has had a parallel effect on pathogen prevalence and ARG abundance. In ring-tailed lemurs,

for example, it was shown that ARGs were in greater abundance in captive populations compared to wild, and ARGs that could impact human health were correlated to the level of human disturbance in the location of varying lemur populations (*Bornbusch & Drea, 2021*). Further, mouse lemurs dwelling in more human-disturbed areas harbored pathogenic bacteria also found in nearby dwelling human, rodents, and livestock (*Bublitz et al., 2015*). It is unknown whether the ARGs identified in wild populations share a similar genetic profile to the profile of the human microbiomes present in the area. Information on general human and lemur interactions, even no interactions, could be informative of the dispersal of reservoirs for transmission.

The landscape of genomic analyses capable of comparing bacterial communities ranges from fast but less sensitive 16S sequencing to highly discriminatory but labor intensive metatranscriptomics (*Knight et al., 2018*). Application of metagenomic sequencing can strike the balance for understudied microbiomes and allow for comparing diversity at the microbial species scale without a priori assumptions of what species should be expected (*Knight et al., 2018*), and capture more gene-level diversity which cannot be evaluated from taxonomy gene marker techniques (*Ranjan et al., 2016*). Although challenges remain for positive identification of rare species from short-read sequencing, it is nevertheless a powerful approach for examining patterns in abundantly present and known species (*Ranjan et al., 2016*), for diagnosis of present pathogens (*Naccache et al., 2014*), and for identifying compositional differences between environmental samples (*Munk et al., 2022*). Additionally, genetic closeness of detected species or genes of interest shared by humans and animals can be used to infer zoonotic transfer and further combined with epidemiological information to identify links between ARG presence, species richness, and risks of illness and transmission. However, the abundance of antibiotic resistance genes may be underestimated, and a latent, or undocumented diversity in established databases, population of ARGs exists within and between different hosts and environments (*Inda-Díaz et al., 2023*). Sequencing metagenomic analysis can help identify the reservoirs of rare or unknown species and offer a starting place for hypothesis generation for complimentary methods, such as functional metagenomics, to fill in the species knowledge gaps (*Zhang et al., 2021*).

Here, we aimed to identify the ARG burden and diversity between humans and wild lemurs near Ranomafana National Park in Madagascar. This unique system provides an opportunity to examine this interplay in low-resource, rural, tropical communities where exceptional biodiversity and human-wildlife overlap create unusually high potential for novel zoonotic events. Comparison of the respective bacterial species and ARG profile lays the foundation for understanding ecological and evolutionary patterns outside of agricultural and clinical settings. This has implications for documenting potential downstream or indirect selection pressure that anthropogenic drug use has on an ecosystem regardless of direct human and animal interaction.

## MATERIALS & METHODS

### Sample collection and demographic survey

As a component of a One Health research platform in Ifanadiana District, Madagascar, a household survey was conducted from June to August 2017 in eight communities in roadless areas <5 km from Ranomafana National Park to collect information regarding household member demographics, antibiotic usage, household illness, exposure to wildlife, and previous illness with diarrheal disease. Within these communities, we have documented diverse global health challenges including high prevalence of enteric infections and resistance genes regardless of antibiotic class, and zoonotic human-wildlife linkages (*Bublitz et al., 2014*; *Bublitz et al., 2015*; *Bodager et al., 2015*; *Zohdy et al., 2015*; *Ragazzo et al., 2018*). Human participation in the study and collection of survey data were approved of and reviewed by the Emory Internal Review Board (IRB00093812). Before survey administration, informed oral consent was gathered and documented. Household members were also asked to voluntarily submit a fecal sample regardless of history of diarrheal illness. Fecal sample IDs were linked to their corresponding household survey responses, and deidentified for downstream analysis. Fecal samples were collected from captured wild brown mouse lemurs (*Microcebus rufus*) along footpaths near the villages. The mouse lemurs were trapped using banana-baited Sherman traps (XLR; Sherman Traps Inc., FL, USA), and set overnight at 16:00 and checked at 05:00. One microliter of fresh fecal samples was collected from individual trapped lemurs by using a sterile tongue depressor and transferring the sample into a cryovial filled with approximately 0.8mL RNAlater. The Emory University Institutional Animal Care and Use Committee provided full approval for this research (#3000417) and the field research procedures were approved by Madagascar's Ministry of Environment, Ecology and Forests (permit nos: 028/17; 083/17; 136/17; 146/17; 164/17).

### DNA extraction and sequencing

DNA was extracted from fecal samples using a standard Zymo Inc. bead-beating kit. Whole metagenome shotgun sequencing was performed on the NextSeq 2000 platform using Illumina DNA library preparations. Sequencing produced separate forward and reverse paired-end fastq files, which served as inputs for bioinformatic processing. All mouse lemur metagenomic samples and select human fecal samples from each of the surveyed eight communities were included for metagenomic sequencing and downstream analyses. Human samples were selected through stratified random sampling per village. To be included for selection, households had to include at least one adult, one school-aged child (5–17 years), and one child under 5 years. Eligible households were grouped by their home village and a random three households were selected within each community, sampling without replacement. From the selected households, a sample was chosen for each age group. If only one household member represented an age category, then their sample was selected. If more than one household member was represented by an age group, then a second random sampling was done to choose the representative sample for that age group.

## Bioinformatic processing and quality control

An overview of the bioinformatic workflow is shown in Fig. S1. Data quality was assessed with FastQC (v.0.11.9) before and after adaptor trimming and removal of host reads (*Andrews et al., 2012*). Read quality trimming was conducted using Kneaddata (v0.10.0) with –*trimmomatic*, which employs Trimmomatic (v0.39-2) (*Bolger, Lohse & Usadel, 2014*) and Bowtie2 (*Langmead & Salzberg, 2012*) to remove adaptor reads and reads mapping to the human genome refence GRCh37 (*Church et al., 2011*), keeping reads at or above Phred 33. Reads from lemur microbiomes were additionally mapped against a draft genome assembly of *Microcebus murinus* (GCF_000165445.2) (*Larsen et al., 2017*), selected for its high level of completeness of assembled chromosomes, representation of male and female chromosomes, and shared ancestry to *M. rufus*. The *M. murinus* assembly was indexed using bowtie2-build. The forward and reverse paired-end reads from lemur microbiomes were mapped to the indexed assembly using Bowtie2 (v2.5.0) and saved as a SAM file. Unmatched reads (and therefore non-host reads) were subsequently removed using SAMtools' sort and fastq functions (*Danecek et al., 2021*). After filtering, only metagenomic sequences with $1 \times 10^6$ reads or more were considered for further analysis or assembly. Metagenomes were assembled using SPAdes (v3.15.4) (*Nurk et al., 2017*) on the trimmed and decontaminated paired-end fastq files using the –*meta* parameter. Separate forward and reverse fastq files were used as input with the -*1* and -*2* flags, respectively. The human DNA-scrubbed analysis sequences are available under the Bioproject PRJNA1008138.

## Classification of bacterial species and ARGs

Taxonomic composition of the filtered reads was first calculated using MetaPhlAn4 (*Blanco-Míguez et al., 2023*), with flags –*input_type fastq*, –*unclassified_estimation*, and –*bowtie2out*, to estimate relative abundance of the both classified and unclassified reads which did not match gene markers in the database. This was followed by a second analysis against the Bowtie2 indices to calculate relative abundance of bacterial-associated reads only using the flags –*input_type bowtie2out*, –*t rel_ab*, –*ignore_eukaryotes*, –*ignore_archaea*. The vOct22 Bowtie2 database available for MetaPhlAn4 was downloaded using the command: *metaphlan –install –bowtie2db* and used as a reference for taxonomic markers. To calculate the abundance of ARGs, we enumerated the reads per kilobase per million (RPKM) relative to the amount of detected bacterial reads in the sample. We derived this formula from *Munk et al. (2022)*, but accounting for reads. The formula is as follows:

$$\frac{\text{Gene reads}}{(\text{Length of gene, kilobases}) \times (\text{Total bacterial reads})} \times 10^9.$$

Filtered reads were first processed through KMA (*Clausen, Aarestrup & Lund, 2018*) using the AMR Finder Plus nucleotide sequence database to identify ARGs and virulence genes (*Feldgarden et al., 2021*). ARGs were specifically subset from virulence genes based on classification from the Bacterial Antimicrobial Reference Genes database for sequences related to antibiotic resistance (PRJNA313047). Unique genes were categorized based on annotated gene symbols and unique alleles were categorized based on sequences matching to present NCBI nucleotide reference sequences. Results were included if the detected allele had a template coverage greater than or equal to 60 percent and a query identity
greater than or equal to 90 percent, and if they had at least three reads assigned. The results file was then joined with a .mapstat file generated by KMA to quantify the number of reads assigned to each reference sequence. To contextualize the reads relative to bacterial content, the filtered fastq files were also run through Kraken2 (*Wood, Lu & Langmead, 2019*) using the flags *–paired, –report, –classified-out,* and *–unclassified-out*, and referencing the Kraken2 Standard database (26 September 2022) to obtain the number of reads rooted at the bacterial level and the number of unclassified reads, or reads unable to be identified using the database classifications, in the sample. RPKM was calculated for each ARG allele.

## Statistical analysis of species and ARG diversity

Final statistical analyses were conducted in R (*R Core Team, 2013*). Continuous values and counts of discrete data were assessed for normal distribution. The Wilcoxon rank sum test in the stats package (v4.0.4) was used to compare human and lemur metagenomes differences in median total detected reads, proportion of reads mapping to higher order taxa, Shannon indices for bacterial species and ARG allele diversity, RPKM of ARG reads mapping to specific antibiotic classes, and median number of species per sample. Alpha and beta diversity metrics were calculated using the vegan package (*Oksanen et al., 2022*). Shannon diversity was calculated based on the presence and absence of detected species or alleles. For this system, two measures of alpha diversity were used to help to capture a better understanding of detected species. The Shannon diversity index allows for comparison of both the richness and evenness of the community structure thus relying on both the abundance and the overall number of species, while Chao1 gives a greater weight to low abundance species present in a sample to help predict the likely number of missing species (*Kim et al., 2017*). Principle component analyses (PCA) were performed on the relative abundances of species and RPKM values of ARG alleles transformed into centered log-ratios to account for the compositional nature of metagenomic data (*Gloor, Macklaim & Fernandes, 2016*). Subsequently, Aitchison distance was calculated to assess between-sample differences in species/allele diversity. Chao1 and rarefaction statistics were calculated using the iNext package using the sum of the presence of each species or allele detected within human or lemur microbiomes as input (*Hsieh, Ma & Chao, 2022*).

## Differential gene abundance analysis

Differential gene abundance between humans and lemurs for antibiotic resistance genes was conducted on the read counts, summarizing the allele hits to the level of the gene using ALDEx2 (v.1.35.0) (*Gloor, Macklaim & Fernandes, 2016*). First, the raw read counts were transformed using the command *aldex.clr(),* with Monte-Carlo sampling set to 128 and the measured denominator set to "all". To account for both composition and scale in the read counts, uncertainty was added to the model using a gamma value of 0.5. A sensitivity analysis for significance of unique features at various values of gamma was conducted for reads summarized at the gene level and per corresponding antibiotic class associated with resistance (See Fig. S3) (*Nixon et al., 2023*). To statistically evaluate the transformed abundances, *aldex.effect()* and *aldex.ttest()* were used to perform Welch's *T*-test and a Wilcoxon rank sum test, and corrected for false discovery using Benjamini–Hochberg corrected *p*-values (<0.05). Final results were plotted using the *aldex.plot()* function.

### Sequence comparison of antibiotic resistance genes shared between humans and lemurs

AMR Finder Plus (*Feldgarden et al., 2021*) was used to identify contigs with ARGs. Contigs with the same ARG detected in at least one human and one lemur microbiome were extracted from their sample assembly using *bedtools getfasta*. These sequences were used to create a custom nucleotide BLAST database (v2.12.0). Each sequence was then queried against the database to identify the pairwise percent identity of each gene compared to other detected genes of the same type. A second BLAST comparison was conducted by extracting the 1,000 base pair regions before and after ARGs commonly present in human and lemur samples, and the pairwise percent identities were quantified. To gain additional insight into the genomic contexts of highly similar ARG-regions, ARG-bearing contigs were extracted from the assemblies and annotated using Bakta (v1.7.0) (*Schwengers et al., 2021*). For ARG-regions in which commonly found ARGs between pairs of subjects had greater than 90% similarity, the gene synteny of the annotated contigs was inspected and visualized using Gggenes (*Wilkins, 2020*).

## RESULTS

### The diversity and abundance of bacterial species and ARGs differ between humans and lemurs

A total of 73 human-derived samples and 15 lemur-derived samples were selected for shotgun metagenome sequencing. Of these samples, 57 human samples had greater than or equal to $1 \times 10^6$ total reads after decontamination of human reads. After mapping to the lemur genome assembly, 11 lemur samples had greater than $1 \times 10^6$ reads for analysis. The metagenomes of these 68 samples were further characterized for bacterial species abundances and presence of antibiotic resistance genes.

There was a higher total number of reads in lemur samples compared to human samples after decontamination (Fig. 1A) which could not be explained by a large number of small reads present. Although both human and lemur read libraries had average read lengths within an acceptable range for downstream mapping and assembly, the average sequence length for lemurs was higher and tightly ranged (147 to 139 reads) ($p < 0.05$) (Fig. 1B). For the majority of human and lemur microbiomes, the most abundant taxa identified belong to kingdom Bacteria (Fig. 1C). However, lemur metagenomes noticeably contained higher relative abundances of reads unable to be classified taxonomically. A comparison of unclassifiable reads to taxonomically classified reads did not show significant differences in GC content, Q30 score, average length, or minimum length (Fig. S2). This suggests that much of the diversity of taxa in lemur microbiomes was not represented even in large database collections used for widescale taxonomic composition classification.

Although different in rank of abundance, human and lemur fecal metagenomes include a high abundance of bacteria from the phylum Bacillota (Firmicutes), and human metagenomes are also dominated by Bacteriodota. Several lemur samples were contrastingly dominated by species from Pseudomonadota (Proteobacteria) (Fig. 2A). Humans and lemurs shared 55 of 452 known bacterial species (12.2%) that had at least

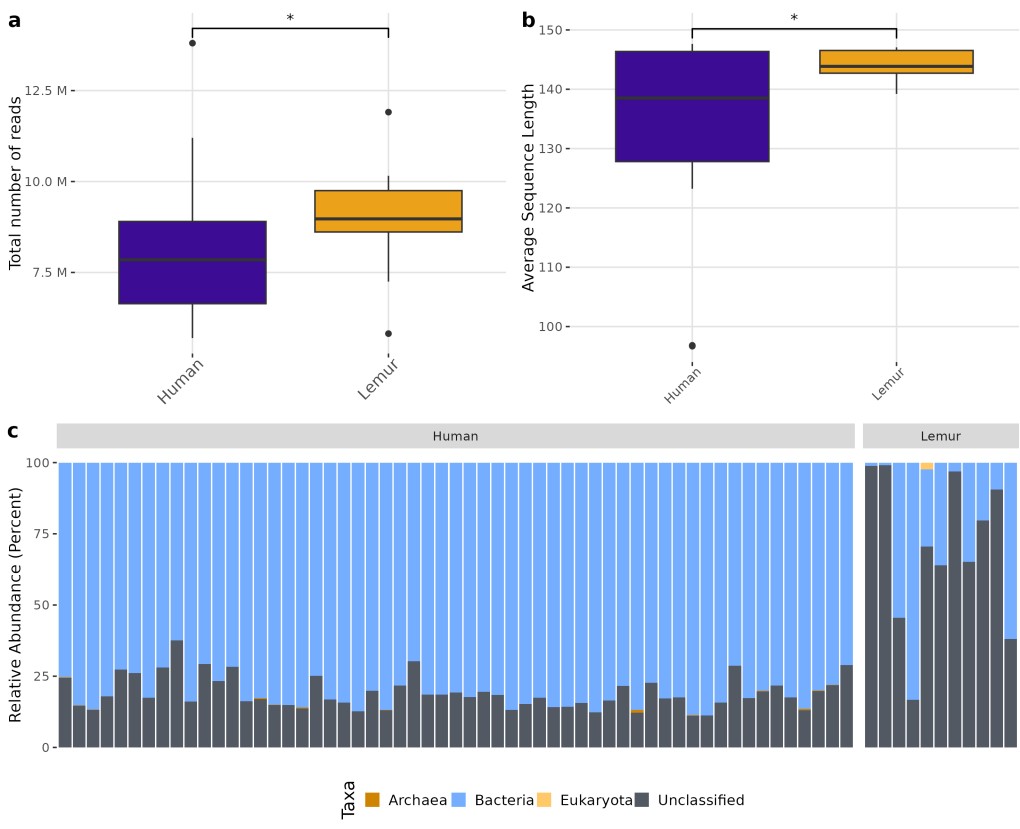

**Figure 1** **Human and lemur metagenomes are different in the classification of sequence reads to higher order taxa.** Sequences were filtered for adaptor sequences, tandem repeats, and reads mapping to human or lemur reference assemblies. (A) The number of total read pairs is significantly higher in lemur fecal metagenomes compared to human fecal metagenomes (Wilcoxon Rank Sum, $p < 0.05$). (B) The average length of reads within the lemur fecal metagenomes is higher than human fecal metagenomes (Wilcoxon Rank Sum, $p < 0.05$). (C) The relative abundances of kingdom-level taxa was quantified using MetaPhlAn4. Reads unable to be identified as belonging to a higher order were designated as "unclassified". In general, humans and lemurs have the majority of the abundance of taxa assigned to Bacteria or Unclassified; however, lemur fecal metagenomes are characteristically higher in the abundance of unclassified reads.

0.01% abundance within a single metagenome and occurred in at least 10% of all samples, suggesting there are many rare species detected in the system. The Shannon index between the two groups was higher in humans ($P < 0.05$) (Fig. 2B). Distinct clustering by sample source was also observed when examining the compositional differences between sample sources. Lemur microbiomes grouped more with other lemur microbiomes without overlapping human microbiomes, and human samples overlapped regardless of the village of residence (Fig. 2C). By Chao1 estimates, when considering detectable species including those in the lowest abundance (<10% of samples) there were fewer species able to be sampled from lemurs (590, 95% confidence interval (CI): 496–729) compared to humans (1091, 95% CI [1057–1144]) (Fig. 2D).

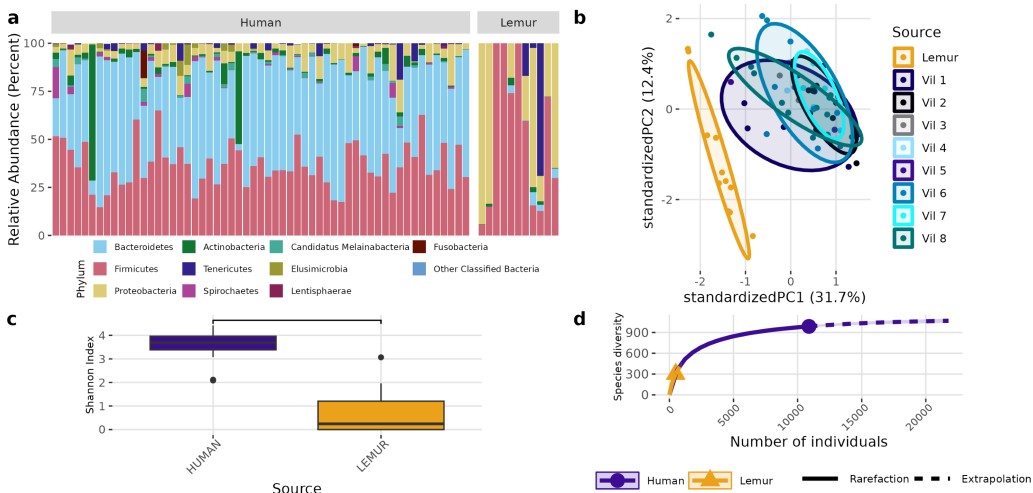

**Figure 2** **Bacterial species communities are distinct between humans and wild lemurs.** (A) Relative abundance of bacterial phyla detected in human and lemur derived metagenomes. (B) Principle component analysis using Aitchison distance on centered log-ratio transformed abundances of individual species stratified by the home village for each human sample or if it was sourced from a lemur, and demonstrates component dissimilarity between human and lemur samples. (C) The Shannon diversity indices of the species detected in human and lemur metagenomes show a difference in the mean value between host sources. Statistical significance was determined by Wilcoxon rank sum analysis at $p < 0.05$. (D) Rarefaction curves of human- and lemur-associated bacterial species, where the $x$-axis is the sampling effort of available individual bacterial species and the $y$-axis is the estimated richness.

A different pattern emerged when examining the overall composition and abundance of ARGs detected in human and lemur metagenomes. We identified reads mapping to 107 unique ARGs, which comprises 217 unique alleles. Two lemur samples had no reads matching ARGs in our database. Individual microbiomes varied in abundance of ARGs grouped by the class of antibiotic to which they confer resistance, but all human microbiomes carried genes associated with tetracycline and trimethoprim, and 55/57 human metagenomes carried resistance genes to beta-lactam antibiotics. In contrast, there was no one shared antibiotic class among detected ARGs in lemur microbiomes, but all classes seen in human microbiomes were represented in at least one lemur microbiome (Fig. 3A). Lemur microbiomes had no statistical difference in ARG richness compared to human microbiomes ($P < 0.05$) (Fig. 3B). Still, lemur microbiomes clustered distinctly in their ARG diversity from human microbiomes, however human microbiome ARG profiles from all resident villages overlapped with one another (Fig. 3C). Concordantly, rarefaction estimates, alongside Chao1 calculations suggested that the estimated maximum number of ARG alleles to be sampled are likely similar for lemurs (201, 95% CI [166–264]) and humans (206, 95% CI [160–302]) (Fig. 3D).

Five antibiotic resistance genes were in significantly greater abundance among human microbiomes compared to the lemur microbiomes (Figs. 4A–4C). These genes consisted of *dfrF* and five separate tetracycline-resistance genes, *tet* (32), *tet* (40), *tet* (W), and *tet* (Q). The effect size of the difference remained significant for these genes when considering the

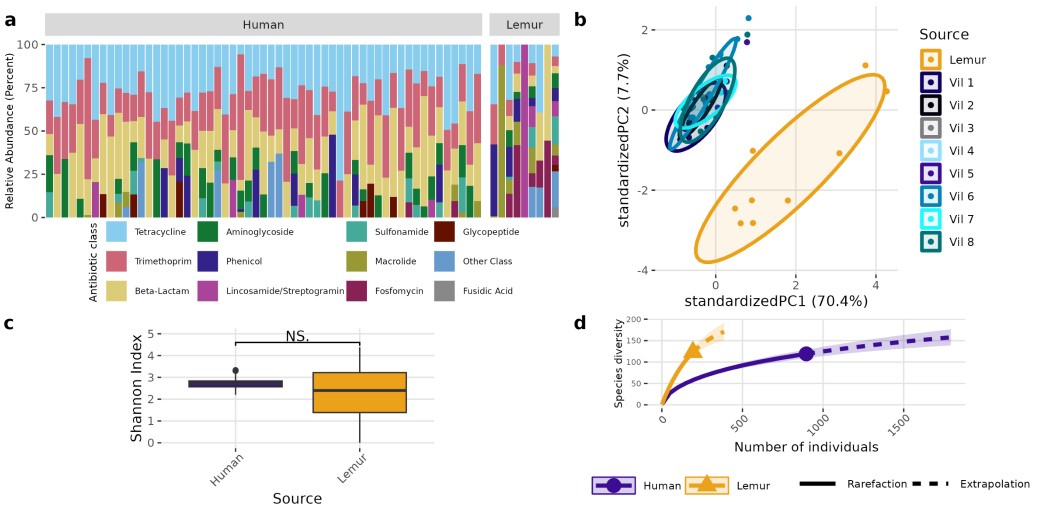

**Figure 3** **Antibiotic resistance gene abundances are distinct between humans and wild lemurs.** (A) Relative abundance of genes by their associated antibiotic resistance classes detected in human and lemur derived metagenomes. (B) Principle component analysis using Aitchison distance on the centered log-ratio tranformed abundances of unique ARGs stratified by the home village for each human sample or if it was sourced from a lemur show a distinct grouping of lemur samples with other lemurs and separate from humans. (C) Shannon diversity index of the unique ARG alleles detected in human and lemur metagenomes and demonstrates a difference in mean values between host source. Statistical significance was determined by Wilcoxon rank sum analysis at $p < 0.05$. (D) Rarefaction curves of human- and lemur-associated ARG alleles, where the $x$-axis is the sampling effort of available alleles and the $y$-axis is the estimated richness.

difference in variance in the data points between the two groups (Fig. 4B). When grouped by associated antibiotic class of resistance, no associated class groups were differentially abundant between humans and lemurs (Fig. S3).

## Humans and lemurs share highly conserved integron-associated ARGs

To capture more specific ARG dynamics between humans and lemurs, we quantified and compared assembled ARGs that were detectable in both human and wild lemur metagenomes. A total of 14 ARGs were detected in common between human metagenomes, with all 57 human metagenomes sharing at least one gene with at least one of three lemur metagenomes.

ARGs of the same type were compared between each metagenome containing that gene. Overall, ARGs from different metagenomes were highly similar, with a median nucleotide sequence identity of 99.51 (98.55–99.79) for human-human, 99.67 (99.24-100) for human-lemur pairs, and 100 (100-100) for lemur-lemur comparisons (Fig. 5A). Similarly high levels of sequence identity were found within each ARG (Fig. 5B). The largest ranges of diversity were among pairwise comparisons of *tet(O)* and *tet(Q)* genes, respectively (Fig. 5B). We also compared the genetic context surrounding the shared ARGs to determine the similarity of genomic context regardless of ARG sequence conservation. Sequence identity of the 1,000 base-pair flanking regions were also nearly identical between

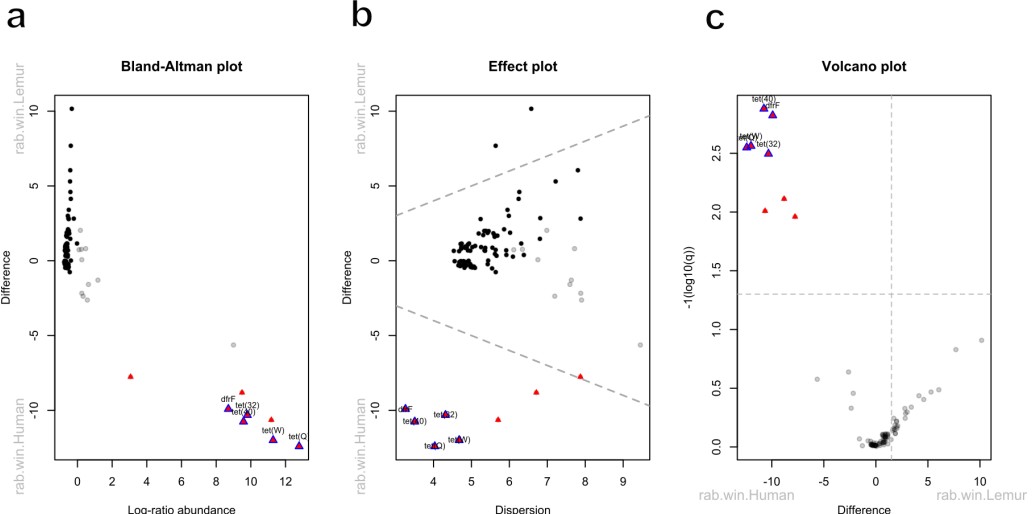

**Figure 4 Antibiotic resistance genes vary in abundance between human and lemur microbiomes.** Raw read counts summarized at the gene level were compared for differential abundance using ALDEx2. Red triangles indicate a significant difference in abundance by an effect value >2. Blue outlining indicates 95% confidence that the value does not intersect zero. Black dots indicate rare and non significant genes while gray dots signify abundant but non-significant genes (A) Bland-Altman plot demonstrating the relationship between the difference between groups in median centered log-ratio (clr) values of each gene and the relative abundance of those genes. (B) An effect plot of the difference between groups in median clr values of each gene and the difference in dispersion, with the dotted lines representing values where dispersion and difference are equal. (C) A volcano plot demonstrating the abundance in the clr values of each gene . The dotted x-intercept line indicates values at a posterior predictive *p*-value of 0.001, and the y-intercept line indicates a 1.5-fold difference in log abundance.

samples. Specifically, the regions around four ARGs (*aadA1, dfrA1, qacEdelta1,* and *sul1*) were from seven human metagenomes and one lemur metagenome showed highly similar pairwise sequence comparison of the human-human and human-lemur source pairs (Fig. 5C). Only one gene, *lsa(D),* had a pairwise identity score less than 100 percent (Fig. 5D).

Because the contexts of these genes were so well conserved, we investigated the gene synteny to understand if there were shared AMR genes in close genetic proximity. ARGs that were co-occurring within assembled contigs included *dfrA1* with *aadA1, sul1,* and *qacEdelta*. Of the eight samples containing *dfrA1*, seven contained a conserved *aadA1* gene and one sample had a broadly categorized *aadA* region (Fig. 6). Five of these samples had a *sul1* downstream of *aadA1*, and three contained *intI1* (encoding class 1 integrase), including the single lemur sample. Of the seven human microbiomes containing a *dfrA1-aadA* pairing, residents were from four different villages, with one pairing from the same household. The single lemur sample was collected closest to a village that none of the human residents carrying this cassette were from.

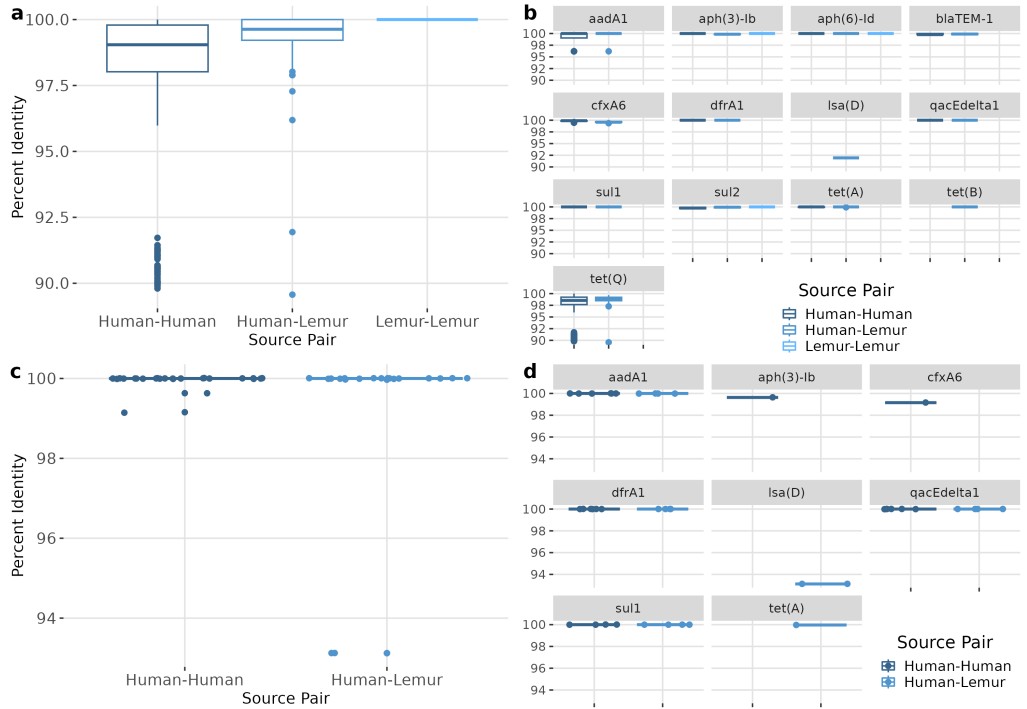

**Figure 5** **ARGs shared between humans and lemurs are highly conserved.** (A) Percent identity for each ARG that was detected in at least one human and one lemur. Pairwise comparisons of the gene source, human or lemur metagenome, was calculated using blastn where positive sequence hits were at least 90% of the query length. (B) The pairwise source comparisons of the percent identity were then stratified by the specific ARG query. (C). Percent identity for the 1,000 base pair region before and after the query gene of interest that was detected in at least one human and one lemur. Pairwise comparisons of the ARG-region source, human or lemur metagenome, was calculated using blastn where positive sequence hits were at least 90% of the query length. (D) The pairwise source comparisons of the percent identity were then stratified by the specific ARG-region query.

## DISCUSSION

To our knowledge, this is the first study to directly compare the antibiotic resistance profiles from human and wild lemur microbiomes from the same geographic space, thus providing further insight into antibiotic resistance gene flow between residential human and wildlife host populations. We quantified and compared the bacterial species and ARG abundances present in human and lemur metagenomes and found their overall profiles to be distinct in both bacterial species and ARG distribution, while human microbiomes from different villages were largely comparable to one another. We also detected some differentially abundant genes among human microbiomes conveying resistance to tetracycline and aminoglycosides. Lastly, we assessed the genomic similarity of ARGs shared between human and lemur microbiomes and found a shared multidrug resistant mobile gene cassette.

Understanding the bacterial composition of microbiomes from hosts within a larger ecological community helps to establish the biological baseline for future surveillance of
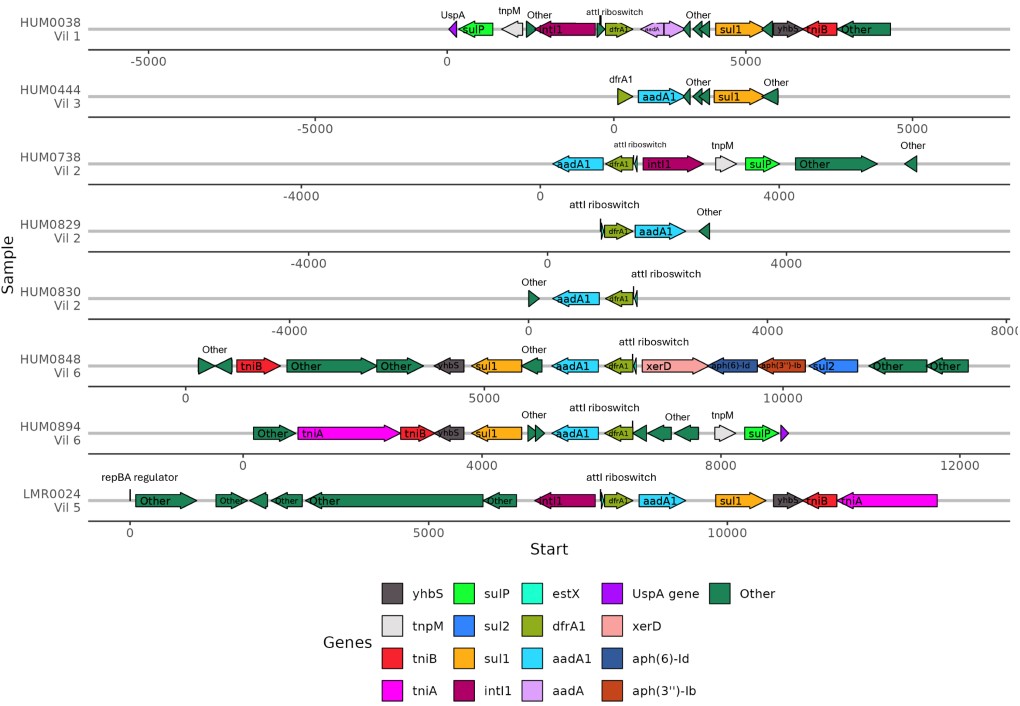

**Figure 6  Human and lemur dfrA1-aadA1 cassette synteny.** Contigs containing highly similar dfrA1 and aadA1 genes were compared from one lemur and four human samples. Sequence annotations were identified using Bakta. Samples from individuals are represented for each line and annotated with the individual and geographic source. Genes present on two or more contigs, or antibiotic resistance-associated genes labeled and represented by colored arrows. Other detected genes unique to the contig are labeled as "other". Sequence coordinates are aligned relative to the present dfrA1 gene on their respective contigs. Arrows indicate the strand direction of the detected gene.

spillover. In this analysis, humans and lemurs were largely distinct in their microbiome species and ARG abundance and distribution. We found that lemur microbiomes were far less rich in known bacterial species compared to human microbiomes despite the quantity of available DNA in the sample. This is likely explained by a limitation in the detectability of uncharacterized species in our chosen database, which also highlights a larger issue of the current state of curated taxonomic databases available for metagenomic analysis. Even with this bias, though, it is reasonable to conclude that the populations from wildlife, being under-sampled across studies, would likely drive this difference even further from humans.

In contrast the Shannon index for ARGs was not different between lemur microbiomes and human microbiomes, and the number of ARGs detected between the two groups were highly similar. Humans and lemurs shared proportionally few types of ARG alleles, but both groups had a similar absolute number of ARG allele types detected. The diversity of alleles suggests less detection bias that would be preferential toward human microbiomes. It is still possible that the full scope of ARGs is yet to be known (*Inda-Díaz et al., 2023*), but in this system there is evidence to suggest that at least what can be known about antibiotic resistance genes is comparable between humans and wildlife. The structure of resistomes within the gut microbiomes of vertebrates outside of humans are influenced

by numerous host-associated factors and environmental factors, including habitat and the threatened status of the wildlife population (*Huang et al., 2022*). For this study, mouse lemurs were sampled along roadways specifically to detect patterns in the resistomes in an area of human and wildlife crossover. The unique life histories and diet of non-human primates from human communities would lead to an expectation that gut bacterial species and present ARGs are likely distinct, as has been demonstrated with comparative analyses of the gut microbiomes of humans and non-human apes (*Amato et al., 2019*). Our study is consistent with this pattern when comparing human to sympatric lemur microbiomes, as the lemur microbiome is largely divergent in species and ARGs present. Nevertheless, the presence of highly conserved ARGs could be the result of shared host traits selecting for specific microbial functions within the gut or from shared lineages acquired from common overlapping environment.

Some antibiotic-resistance genes were more abundant among humans, though resistance to no one class was more abundant. It is notable that four of the five differentially abundant genes belonged to tetracycline-resistance genes and one aminoglycoside-resistance genes. Phenicol and tetracycline class drugs have been used extensively in agriculture (*Roberts & Schwarz, 2016*) and thus could end up trickling into natural settings, impacting how often wildlife become exposed to these ARGs compared to humans. The synergy of clinical and agricultural use could explain why there are overlaps in top abundance. For example, Since 2009, the World Health Organization has recommended that sulfamethoxazole + trimethoprim, doxycycline, or tetracycline be used as first-line choices for pre- and postexposure treatment to *Yersinia pestis*, the pathogen causing plague and which is endemic in Madagascar and responsible for periodic large outbreaks, including during 2017 (*Nguyen, Parra-Rojas & Hernandez-Vargas, 2018*; *World Health Organization, 2021*). The diversity of region-specific usages of antibiotics suggests that there is likely no single pressure resulting in the maintenance of the most abundant ARGs, but it does call for a One Health awareness toward the stewardship of different classes so that these drugs can remain effective for interventions, such as management of plague.

Fourteen assembled ARGs were shared between humans and lemurs, though there were 69 distinct ARGs among assembled metagenomes. Presence of shared genes is a signifier of potential ARG reservoirs for human and agricultural pathogens. Among the shared ARGs, several have been detected in pathogen samples with phenotypic resistance to their corresponding antibiotic class, including *aph(3″)-Ib* (*Ojdana et al., 2018*), *aph(6)-Id* (*Ikhimiukor et al., 2022*), *qacEdelta1* (*Chen et al., 2023*), and *cfxA6* (*Binta & Patel, 2016*). *DfrA1*, *aadA1*, *aph(3″)-Ib, aph(6)-Id*, all have a high risk of contributing currently or in the future to pathogen multidrug resistance (*Zhang et al., 2021*). The *lsa(D)* gene, responsible for lincosamide resistance, was detected in diseased farm-raised fish and attributed to emerging fish pathogens (*Shi et al., 2021*). We did not identify a clear village-level association between lemurs and humans that had these shared genes. Therefore, the high similarity could be explained by strong selective pressures within the environment to conserve these gene structures, or it could be explained by ongoing drift of bacteria horboring these genes moving between human and lemur populations *via* uncharacterized pathways, such as river systems or intermediary contact between wildlife and other domestic

or peri-domestic animals. Many of these genes have a prevalence in other global areas where genetic sequences of ARGs sourced from different metagenomes are also highly conserved (*Osei Sekyere & Reta, 2020*). For either scenario, detection of clinically significant ARGs in wild lemur populations that may not be directly interacting with human communities signifies just how diffuse the community resistome is and may make combating drug resistance more difficult as human and wildlife are brought more and more into contact.

We did identify a common class 1 integron in close genomic context with multiple drug resistance genes present in several human and one lemur microbiome. Common characteristics of a class 1 integron are encoding of intI1 at the 5′ coding end, followed by with a variable cassette region and then encoding of *qacEdelta* and *sul1* at 3′ coding sequence (*Deng et al., 2015*). Other globally distributed gene cassettes harboring trimethoprim-resistant *dfrA* and aminoglycoside-resistant *aadA* genes are known to be associated with class 1 integrons (*Rakotonirina et al., 2013*; *Adelowo et al., 2018*; *Chaturvedi et al., 2021*). Specifically, these gene cassettes have also been found in known patient samples in Madagascar's capital Antananarivo among ESBL-producing Enterobacteriaceae, with the most frequent cassette pairing being *drfA17-aadA5* (*Rakotonirina et al., 2013*). Contrastingly, we did not identify this specific cassette among any of the microbiomes under consideration in this study from our rural community members. The *dfrA1-aadA1* cassette among our study samples is dispersed between several members of different villages, though more investigation is necessary to understand if its prevalence is hallmark of the specific region. Class 1 integrons harboring *dfrA1* can move between species of gram negative organisms *in vivo* (*Van Essen-Zandbergen et al., 2009*). In the context of our study and the growing body of evidence that human-driven antibiotic use drives higher antibiotic resistance profiles in animals and in wildlife, we should be concerned that even non-agriculture animals are maintaining highly similar ARGs to humans in their microbiomes. Stewardship efforts necessary for this system may focus on closing off pathways between human-developed space and wildlife and conservative use in agriculture. Detection of ARGs through metagenomics or other screening is a useful tool for increased surveillance efforts, but it will take additional research to develop meaningful relationships between genetic abundance and the frequency of spread between species or changes in phenotypic resistance to antibiotics. Optimistically, the advent of technologies such as long-read sequencing offer a compliment to identifying species genomes directly as sequences and as reference scaffolds for short read sequences. Given that antibiotic resistance is often a trait maintained when bacteria are consistently exposed to antibacterial chemicals, more work must be done to monitor whether individual genes are continually being reintroduced to wildlife metagenomes from humans to better understand how stable the lemur metagenome niche is as an ARG reservoir.

## CONCLUSIONS

In this study, we took a metagenomic approach to characterize ARG presence in a specific ecological system and uncover previously unexplored comparisons between humans and wildlife. Our observations add to the growing effort to characterize the global extent of

ARG presence, the range of which is still limited especially in lower- and middle-income countries. These findings reflect some known global patterns of drug resistance prevalence and highlight unique patterns for this geographic area. This research supports a continued effort to monitor antibiotic usage for humans and in agriculture, especially effects on non-pathogen members of microbiomes, and their further dissemination into the ecosystem. As more research reveals the extent of ARG transmission through an environment, it is evident that there is an increased needed to investigate intermediary processes beyond individual players' proximities to one another that can lead to drug resistant gene movement through ecological space.

## ACKNOWLEDGEMENTS

For logistical and infrastructural support, the authors thank MICET, Centre ValBio, Benjamin Andriamihaja, Pascal Rabeson, Jean Claude Razafimahaimodison, Jean de Dieu Ramanantsoa, Maya Moore, Jesse McKinney, Telo Albert, Victor Tombotiana Aimé, Emilie Redwood, Katherine Noble, Ria Ghai, Wenjie Mei, Joel Hartter, Karen Bailey, Heritiana Anne Louisette, Rasolondraibe, Amelie Marcelle, Solo Justin, and François Zakamanana. The authors also thank Emily Wissel for critical insights and recommendations for metagenomic techniques. The manuscript's contents are solely the responsibility of the authors and do not necessarily represent the official views of the Centers for Disease Control and Prevention.

### Funding

This research was supported by the Herrnstein Family Foundation, the Marcus Foundation, the Emory Global Health Institute, and a University Research Committee - Halle Institute for Global Research Award. In addition, Brooke M. Talbot and Timothy D. Read were supported by the Office of Advanced Molecular Detection, Centers for Disease Control and Prevention Cooperative Agreement Number CK22-2204 through contract 40500-050-23234506 from the Georgia Department of Public Health. The funders had no role in study design, data collection and analysis, decision to publish, or preparation of the manuscript.

### Grant Disclosures

The following grant information was disclosed by the authors:
The Herrnstein Family Foundation.
The Emory Global Health Institute.
A University Research Committee - Halle Institute for Global Research Award.
the Office of Advanced Molecular Detection, Centers for Disease Control and Prevention Cooperative: 40500-050-23234506.

### Competing Interests

Timothy D. Read and Patricia C. Wright are Academic Editors for PeerJ.

## Author Contributions

- Brooke M. Talbot conceived and designed the experiments, performed the experiments, analyzed the data, prepared figures and/or tables, authored or reviewed drafts of the article, and approved the final draft.
- Julie A. Clennon performed the experiments, authored or reviewed drafts of the article, and approved the final draft.
- Miarintsoa Fara Nantenaina Rakotoarison performed the experiments, authored or reviewed drafts of the article, and approved the final draft.
- Lydia Rautman performed the experiments, authored or reviewed drafts of the article, and approved the final draft.
- Sarah Durry performed the experiments, authored or reviewed drafts of the article, and approved the final draft.
- Leo J. Ragazzo performed the experiments, authored or reviewed drafts of the article, and approved the final draft.
- Patricia C. Wright performed the experiments, authored or reviewed drafts of the article, and approved the final draft.
- Thomas R. Gillespie conceived and designed the experiments, performed the experiments, authored or reviewed drafts of the article, and approved the final draft.
- Timothy D. Read conceived and designed the experiments, authored or reviewed drafts of the article, and approved the final draft.

## Human Ethics

The following information was supplied relating to ethical approvals (*i.e.*, approving body and any reference numbers):

The Emory University Institutional Review Board granted approval to carry out this study (reference number IRB00093812)

## Animal Ethics

The following information was supplied relating to ethical approvals (*i.e.*, approving body and any reference numbers):

The Emory University Institutional Animal Care and Use Committee provided full approval for this research (#3000417)

## Field Study Permissions

The following information was supplied relating to field study approvals (*i.e.*, approving body and any reference numbers):

Field research procedures were approved by Madagascar's Ministry of Environment, Ecology and Forests (permit nos: 028/17; 083/17; 136/17; 146/17; 164/17).

## DNA Deposition

The following information was supplied regarding the deposition of DNA sequences:

The metagenomic sequence reads for this project are available in the Sequence Read Archive: ID PRJNA1008138.
## Data Availability

The analysis code is available on GitHub and Zenodo:

- https://github.com/bmtalbot/Humans_and_Lemurs_2017

*Talbot (2024)*.

The accompanying data are available on Zenodo: Talbot, B., Clennon, J., Rakotoarison, F., Rautman, L., Durry, S., Ragazzo, L., Wright, P., Gillespie, T., & Read, T. (2023). Datasets for Metagenome-wide characterization of shared antimicrobial resistance genes in sympatric people and lemurs in rural Madagascar (1.0.0) [Data set]. Zenodo. https://doi.org/10.5281/zenodo.10402843.

## Supplemental Information

Supplemental information for this article can be found online at http://dx.doi.org/10.7717/peerj.17805#supplemental-information.

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
