# Peer review of "Metagenome-wide characterization of shared antimicrobial resistance genes in sympatric people and lemurs in rural Madagascar"

_PeerJ, doi:10.7717/peerj.17805_

## Round 0.1 · original submission · Major Revisions

Specifically, the reviewers have significant on the description of the study design, improved description of the bioinformatics methods, MGE/integron assay interpretation, and suggested a revisit of the statistical analyses of the microbial communities, switching some description of methods in the discussion to the results, and changing parts of table 1, 2, and 3 to figures.

·

Basic reporting

The authors investigated the pattern similarities between the bacteriome and ARG content of sympatric human and lemur feces in a region of Madagascar. The topic is interesting, but I believe a fundamental reanalysis and rewriting are needed before the manuscript can be considered for publication.
As far as I can guess, the English is good, but in some places it deviates from the correct professional language, e.g. "unidentified". The manuscript is well structured, the references are correct and relevant. However, there are still typographical errors in several places: 'individuals(Parisi et al.)' and sentences without an ending dot. The readability of all these figures needs improvement, perhaps it would be better if they were inserted in vector format or at a much higher resolution. Most of the sequencing data can be accessed using the BioProject identifier provided.

Experimental design

I think a fundamental problem is how to relate human and animal outcomes. At first glance, it seems to assume that the habitats of animals and humans overlap in some way and that this can form the basis for assessing similarities and differences. However, Table 3 shows that only 10 of the 57 households examined have a lemur relationship.
The stratified random sampling mentioned for the selection of the human samples is very oversimplified and should be more detailed and precise.
Throughout the manuscript, the sample sizes are varied, e.g., 57+11 in Table 1, 57+10 in Table 2, and 65 BioSamples under the identifier PRJNA1008138 on the NCBI SRA.
In the description of bioinformatics methods, the authors omit the parameterization of each tool, thus limiting the reproducibility of the study.
And the analysis process cannot be followed, e.g., quality control is preceded by quality filtering.

Validity of the findings

I believe that the most critical point in the whole manuscript, which may concern all ARG results, is the extremely low thresholds used to filter ARG findings. Of course, this is true if I understand correctly what is written: "Results were included if the detected allele had a template coverage greater than or equal to 50 and a query identity greater than or equal to 20." In this thread, 50% coverage is also lower than the commonly used 60% minimum, but 20% sequential identity is well below the widely accepted 90%. Under such mild conditions, the false positive rate is elevated.
The statistical methodology also needs to be reviewed and revised. For instance, the count data cannot be treated as simple ratios due to the different library sizes. They should be analyzed using methods used in the field (e.g., negative binomial model).
Other problematic parts: Bray-Curtis is not an analysis but a distance metric; if it said "We derived this formula from Munk et al. (Munk et al., 2022)", then the formula should be provided.
The methodology section does not describe why and on which samples the PCR was performed.
Similarly, the methodology of the MGE assay is not described here, but some of its details are mentioned in the results. However, what is described in the results does not support mobility, it only identifies very short conserved sequences. This would require prophage, iMGE, or plasmid detection.
It will make sense to comment in detail on the results and the discussion when I can see the results of the analyses carried out on the basis of the above.

Reviewer 2 ·

Basic reporting

The work is very interesting. The manuscript is well structured and clearly presented. Suggest further improve writing, discuss limitation of the study in the discussion part, and improve data presentation (use figure or table), justify the methods like why use RPKM which for me does not have any biological meaning, Below are some specific comments:

The authors need to reconsider the introduction part at Line94-106. I agree the metagenome can provide a more comprehensive picture than those traditional methods. However, its resolution so far is still a bottleneck. I do not think species level analysis is reliable based on short-read metagenome alone. Moreover, I did not get Line102-106, as metagenomics also relies on reference database (only those ARGs in database can be detected), unless using functional metagenomics. The authors need to do more work to review the good and bad of metagenomics.

Line55 what is ARG? Should be defined at the first place.

L225 what is average distance?

L237-238 suggest rephrase the sentence.

L262 suggest to replace MGE by integron since the authors only studied the integron related elements like intI1. The current title is bit misleading.

L270-271 suggest rephrase the sentence.

L313-318 I do not think the explanation of Shannon and Chao1 belong to discussion part.

Suggest change Table 1, Table 2 and Table 3 to figure (or part of the tables to figure) which is better for presentation and easier for understanding.

Experimental design

no comment

Validity of the findings

no comment

Additional comments

no comment

---

## Round 0.2 · Minor Revisions

I believe most reviewer concerns have been adequately addressed. However, Reviewer 1 still has some significant concerns with the experimental design, explanation of methodology, clarity of the results, and has observed that the track changed and final pdf differ. I suggest careful consideration of the relationship of ARG filtering criteria and the revised results, as observed by Reviewer 1. Also, please correct the order of figures. Thank you.

·

Basic reporting

Despite these suggestions, the manuscript has not become any clearer or more verifiable.

Experimental design

No, the test question has not yet matched the results achieved. The methodology is still unclear even in topics that were required in the first round.

Validity of the findings

The conclusions are completely loose, in fact, they could have been drawn without any studies.

Reviewer 2 ·

Basic reporting

no comment

Experimental design

no comment

Validity of the findings

no comment

Additional comments

The order of figures is wrong which has to be corrected before publication.

·

Basic reporting

The authors have effectively responded to the previous reviewers' comments.

Experimental design

The experiment design is appropriate, as are the methods used.

Validity of the findings

The conclusions are, after revisions, well stated, and the link to the original research question.

Additional comments

The original reviewers did an excellent job with their comments and suggestions for improvement, as did the authors' responses.

---

## Round 0.3 · accepted · Accept

Reviews have been adequately addressed.